# Agronomic Effects of *Tectona grandis* Biochar from Wood Residues on the Growth of Young *Cedrela odorata* Plants in a Nursery

**Arantxa Rodríguez Solís** , **Yorleny Badilla Valverde** and **Róger Moya ***

Instituto Tecnológico de Costa Rica, Escuela de Ingeniería Forestal, P.O. Box 159–7050, Cartago 30101, Costa Rica; arantxarodriguezs@gmail.com (A.R.S.); ybadilla@itcr.ac.cr (Y.B.V.)
* Correspondence: rmoya@itcr.ac.cr; Tel.: +506-2550-9092

**Abstract:** Biochar from agroforestry biomass residues is an alternative source of fertilizers for improving the soil fertility. In Costa Rica, *Cedrela odorata* is planted in pure plantations and agroforestry systems and different types of substrates are used in the nursery to enhance the growth and performance of the young saplings. The main objective of the present study was to evaluate the growth (in height, diameter, biomass) and distribution of carbon (C), hydrogen (H), and nitrogen (N) for *C. odorata* plants in a nursery with different application rates of biochar produced from *Tectona grandis* wood residues. The results showed that the above-measured variables were statistically affected by different application rates of the biochar. The stem diameter, total height, number of leaves, and height increment were statistically significantly higher in the substrate with an application rate of 25–50 tons/ha, in contrast to the 0 and 75 tons/ha application rates, which were statistically the lowest. As for the levels of C, H, and N, there were differences, with the highest values of N and C being in the leaves and stem with the 50 tons/ha application rate and the highest values of H for the 75 tons/ha application rate. The above results showed that applying biochar obtained from *T. grandis* residues improved soil conditions, resulting in better growth of *C. odorata* saplings with an application rate of 25 and 50 tons/ha.

**Keywords:** charcoal; plantation species; seeding; soil properties; substrates; soil fertility

## 1. Introduction

Charcoal is a product derived from biomass from agroforestry or urban organic residues, which are used as raw material to be subjected to a pyrolysis process [1]. The biochar from wood biomass by pyrolytic processes can produce a surface area and size and distribution of porosity in adequate soil adsorbent material and other physicochemical properties [2]. These characteristics of the charcoal permits the material to be used as biochar as an alternative to improve the fertility [3] and nutrient retention in the soil [4]. Biochar also improves nutrient cycling and increases fertilizer efficiency, reduces leaching, and stimulates soil microbial activity and plant growth and crop productivity [5].

Biochar also improves certain functions in the soil because of its addition to the soil and brings better changes in the chemical and physical conditions of soil [2]. However, the improvement in conditions depends on the interaction of biochar with the physicochemical characteristics of the soil, climatic conditions, and the rate of its application [6]. It has an immediate effect of increasing the availability of elements, such as potassium, phosphorus, and zinc, as well as calcium and copper to a lesser extent [7,8].

An important element for fast growth of the plants is the substrate used in the nursery [9]. The substrate provides support to the plants, keeping the roots stationary, maintaining moisture, and providing drainage to excess water and oxygen exchange in the root zone [10], and it can even provide nutrients [11].

The advantages of the use of biochar in soil have been shown in different agricultural crops. For example, the application rates of biochar obtained from various raw materials,

such as corn cobs, coffee husks, and shrimp exoskeletons, on the germination and growth process of *Capparis scabrida* Kunth saplings in the nursery stage showed a reduction in germination time and an increase in the leaf biomass [12]. Another study conducted in Peru evaluated the effect of biochar from organic residues and showed a positive effect on the germination of *Ceiba trichistandra* (A.Gray) Bakh. [13].

On the contrary, a mixture of biochar in the soil substrate also ensures the plant growth conditions [14,15]. It improves aeration in the soil by the addition of biochar with different particle sizes [16], which results in greater oxygenation of the root system and the evacuation of carbon dioxide gas produced by roots and microorganisms [15]. Likewise, a mixture of biochar with soil in the substrate also beneficiated plant protection against pests [17]. The growth of beneficial inoculums for plants requires the use of substrates since they are an effective medium for the proliferation and conservation of the fungal potential [18].

*Cedrela odorata* L. is a tree belonging to the Meliaceae family [19]. The wood of *C. odorata* is used in furniture, doors, musical instruments, cabinets and boxes for storage fabrication, and to decorate interiors and boats [20]. The wood of *C. odorata* presents a specific gravity lower than 0.5, and recent studies have shown that trees grown under agroforestry systems (AFS) conditions or in a pure plantation produce wood with good properties and do not present problems during the different wood industrialization processes [21].

This species is massively propagated in nurseries through rapid procedures with different substrates [22], which achieves different effects on plant growth. In Mexico, a study was conducted on the rooting of mini cuttings of *Cedrela odorata* and the effect of perlite and a 2:1:1 mixture of peat, perlite, and vermiculite as substrates for rooting [23], where they found higher survival and rooting percentages when perlite was used as the substrate. In another study, a positive effect of biofertilizer on *C. odorata* plants with *Rhizophagus intraradices* (N.C. Schenck & G.S. Sm.) C. Walker & A. Schüßler and *Azospirillum brasilense* Tarrand, Krieg & Döbereiner was demonstrated in a nursery, where a substrate of soil and washed river sand in a 1:1 ratio and bovine manure at 10% of the volume was used. This study determined that the combination of both biofertilizers in the substrate mixture promoted greater development in *C. odorata* plants [22].

Due to the importance of the substrate in the initial stage of plant growth and development [24] and the great potential of *Cedrela odorata* in agroforestry systems [21], the present work aimed to evaluate the effects of biochar obtained from *Tectona grandis* L. wood in different application rates of the substrate on height, diameter, and biomass under dry conditions and the quality of *C. odorata* plants in the nursery stage. Hence, this trial assessed biochar from *T. grandis* wood as an alternative option to enhance the growth of the *C. odorata* plant, when combined with agricultural soil in a nursey.

## 2. Materials and Methods

This study was carried out in the greenhouse of the Instituto Tecnológico de Costa Rica, Campus Tecnológico Cartago, located in the province of Cartago, Costa Rica (9°51′08″ N latitude and 83°54′31″ W Longitude). It is located at an altitude of 1360 masl, with an average annual rainfall of 2300 mm and a temperature ranging from 17 °C to 24 °C [25], with a Very Humid Forest life zone [26].

### 2.1. Experimental Design of the Trial

The effect of biochar on the growth of *C. odorata* plants was evaluated using a completely randomized design, composed of four biochar application rates, with three replicates. The biochar application rates were 0, 25, 50, and 75 ton/ha of *T. grandis* biochar and control (0 ton/ha), corresponding to the soil without biochar. A PVC black seed tray with dimensions of 31.5 (width) and 52.5 cm (length) with 60 cells of 132 $cm^3$ with inverted quadrangular pyramids, where the top of the part present is 6 cm × 6 cm and the lower part is 2 cm × 2 cm and the depth is 11 cm, were used. The trays were placed in the greenhouse for 42 days. Each application rate was established in three trays, which corresponded to

the replicates of the trial. In each tray, 16 seedlings were placed uniformly. The four plants in the central part were used as the measurement plot.

### 2.2. Substrate Preparation

For each application rate, biochar of *T. grandis* wood produced using the earth pit method was utilized. Extensive details of the biochar production and its characteristics are explained in detail in Berrocal-Méndez [27]. The biochar was ground in an industrial mill to obtain particles ranging from 1–4 mm in size. The soil extracted from the Cartago-Costa Rica area, specifically the A horizon (0–30 cm), had a bulk density of 0.82 (g cm$^{-3}$) and it was sieved with a 1.5 cm mesh.

### 2.3. Preparation of Treatments

Four biochar application rates mixed with the soil were used with the following proportions: control, 25, 50, and 75 tons/hectare. For the calculation of the application rate, a bulk density of the soil of 0.82 g cm$^{-3}$ and a depth of 14.5 cm were taken into consideration. An application rate per plant in the biochar substrate of 0.53 g, 1.00 g, and 1.69 g per 37 g of soil for the four biochar application rates was determined.

### 2.4. Plant Material

Twelve-week-old plants of *C. odorata* raised from seeds obtained from the Pérez Zeledón area were germinated in the GENFORES greenhouse at the Instituto Tecnológico de Costa Rica, San Carlos, Costa Rica, and were used to carry out the experiment. The plants were transplanted to 36 mm pellets after two weeks of seed germination. At the establishment of the trial, the plants were extracted from the pellet and root pruning was performed to eliminate defects in the root system.

### 2.5. Management of the Trial

Eight days after the trial was established, a dose of 50 mL/20 L of foliar fertilizer containing 10.2% total nitrogen, 0.6% ammoniacal nitrogen, 9.5% organic nitrogen, and 50.5% total amino acids was applied. During the entire duration of the trial, fertigation was applied when the amount of water in the substrate was at 50% of its capacity. For this, the total weight of the substrate and the amount of water was quantified, which corresponded to 67.46% and 32.54%, respectively. When each tray was at 83.73% of its total weight, fertigation was applied.

Fertigation consisted of the application of 13.0 g of $MgSO_4$, 10.7 g of $KH_2PO_4$, 2.6 g of $NH_4H_2PO_4$, 13.9 g of $NH_4NO_3$, 4.1 g of urea ($CH_4N_2O$), and 14.1 g of Ca $(NO_3)_2$. In the fertigation application cycle, corresponding to the application of water, all salts were supplied except $Ca(NO_3)_2$, before $Ca(NO_3)_2$ was administered. This fertigation cycle was repeated until the last measurement day of the trial so that two complete application cycles were carried out.

### 2.6. Plant Growth in Height, Diameter, and Number of Leaves

The growth variables evaluated were collar diameter at one mm from the substrate surface, total height, and number of leaves at 7, 14, 21, 21, 28, 35, and 42 days in the four plants of the effective measurement plot. The diameter was measured at the base of the stem and the height was measured from the base of the plant to the height of the apex. The number of leaves was counted per unit.

### 2.7. Plant Biomass

The biomass corresponded to the weights obtained in a dry condition for leaves, stems, and roots, for which samples consisting of three plants per tray (treatment) were used. The total biomass agreed with the sum of the biomass in the leaves, stems, and roots. At the end of the experiment (i.e., after 42 days), the plants were carefully uprooted from trays and roots were washed with water, taking care not to lose fine roots, and

subsequently, stems and leaves were separated from the saplings. Each part was placed in an aluminum container and placed in a drying oven BLUE-M model POM-256C-1 (Electrico Campanic, IL, USA) for 48 h at 50 °C, after which they were weighed to obtain the dry weight (dry condition).

The carbon (C), hydrogen (H), nitrogen (N), and sulfur (S) contents and the C/N and C/H ratios were also determined too. For this analysis, samples were taken from the dried leaves, stems, and roots. Each sample was ground separately and 4–5 mg were taken to obtain three subsamples or replicates per leaf, stem, and root, which were taken in tin capsules to the elemental analyzer (Elementar Vario Cube, Munich, Germany), which determined the C, H, N, and S contents.

### 2.8. Data Processing and Statistical Analysis

Microsoft Excel 365 was used for data preprocessing. The assumptions of normality of the data, homogeneity of variances, and analysis of variance were checked. The source of variation in the analysis was the biochar application rate on the variables of neck diameter, total height, number of leaves, dry biomass, and C, H, S, and N contents in the leaves, stems, and roots at 42 days to determine the effects produced by the different application rates evaluated with the GLM procedure of the SAS statistical program (SAS Institute, 1996, Cary, NC, USA). The differences in the means between treatments were established by employing the Dunnett's test ($p < 0.05$), working with a degree of significance of 95% confidence.

## 3. Results

### 3.1. Growth in Diameter, Height, and Number of Leaves over Time

Figure 1 shows the variation in the stem diameter, height, and number of leaves with time. In diameter, the treatments with the highest values were those obtained for the application rate of 25 and 50 ton/ha, with the application rate of 50 ton/ha being the highest (Figure 1a). Regarding the plant height, it was observed that the three application rates evaluated presented significant differences with respect to the control in the different days evaluated, but some differences can be observed among the application rates (Figure 1b). It is possible to observe that the application rate of 25 ton/ha presented the highest values, followed by the application rate of 50 ton/ha, and finally, the application rate of 75 ton/ha (Figure 1b). Finally, for the number of leaves, it was observed that in the first 35 days of evaluation, the application rate of 25 ton/ha showed the highest values, followed by the application rates of 50 and 75 ton/ha, respectively (Figure 1c). For the 42 day evaluation of this same parameter, the application rate of 50 ton/ha showed the highest value, followed by the application rate of 25 ton/ha, and finally, 75 ton/ha.

### 3.2. Average Growth in the Diameter, Height, and Number of Leaves at 42 Days of Evaluation

Table 1 shows the results of the analysis of variance for these variables on the different measurement days. The diameter of 42-day-old plants after the application rates of 50 and 25 ton/ha presented a higher stem diameter as compared to the control while the application of 75 ton/ha of biochar showed no differences if compared with the control (Figure 2a) saplings. On the contrary, the total height and number of leaves were affected by the quantity of biochar application. The highest diameter and number of leaves was observed in saplings that were supplied 75 ton/ha, followed by 50 and 75 ton/ha, and the lowest values in height were noted in the control samples (Figure 2b). It is evident that all the biochar application rates increased the number of leaves (Figure 2c).

### 3.3. Variation of the Increase in the Diameter, Height, and Number of Leaves

Figure 3 shows the variation in the increase in the height, diameter, and number of leaves with time. Regarding the variation in the increase in the stem diameter and the number of leaves per individual saplings, it was found that the values ranged from 0.1 to 0.3 mm for diameter and from −0.25 to 0.6 for the number of leaves (Figure 3). However, no significant differences were found (Table 1 and Figure 3a,b) in the control samples. In

terms of the increase of height, significant differences were obtained for 14- and 42-day-old saplings (Figure 3c). At seven days old, the increase in height at the 25 ton/ha application rate presented the lowest values. However, in 35-day-old individuals, the three treatments showed significant differences, with a greater increase in height in the 75 ton/ha treatment, followed by the 50 and 25 ton/ha treatments, respectively (Figure 3).

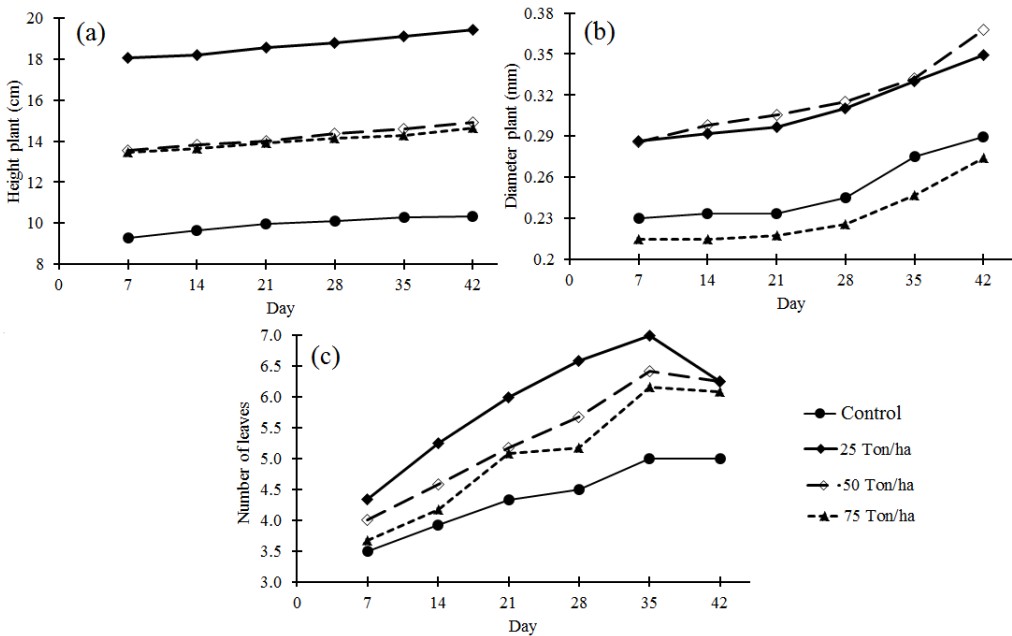

**Figure 1.** Variation of the diameter (**a**), height (**b**), and number of leaves (**c**) of plants of *C. odorata* with different application rates of biochar.

**Table 1.** F values obtained from ANOVA for different parameters evaluated for different application rates of biochar.

| Time | Diameter | Total Height | Number of Leaves | Increment of Diameter | Increment of Height | Increment of Number of Leaves |
|------|----------|--------------|------------------|-----------------------|---------------------|-------------------------------|
| 7 | 7.88 *** | 36.07 | 2.38 *** | 0 | 0 | 0 |
| 14 | 10.57 *** | 34.49 *** | 5.02 *** | 1.07 | 1.07 *** | 3.04 |
| 21 | 12.85 *** | 32.31 *** | 4.61 *** | 1.75 | 0.89 | 1.63 |
| 28 | 13.52 *** | 32.42 *** | 8.79 *** | 0.33 | 2.45 | 1.93 |
| 35 | 10.58 *** | 33.83 *** | 9.65 *** | 0.39 | 1.07 | 2.59 |
| 42 | 9.90 *** | 37.25 *** | 4.63 *** | 1.63 | 6.20 *** | 2.12 |
| **ANOVA for Biomass** | | | | | | |
| | Dry biomass leaves | | Dry biomass in stem | | Dry biomass in root | Total dry biomass |
| 42 | 16.26 *** | | 1.18 | | 2.20 | 3.00 |

Legend: $p < 0.05$, there are significant differences ***.

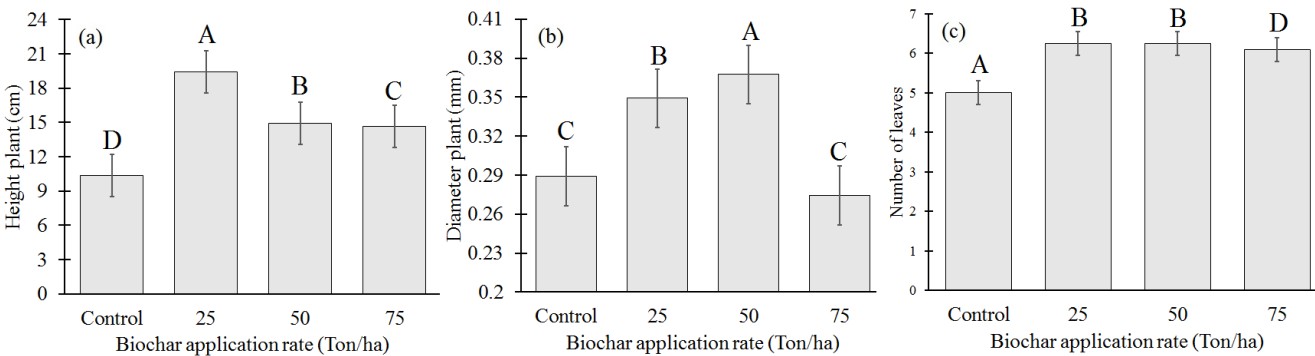

**Figure 2.** Diameter (**a**), height (**b**), and number of leaves (**c**) of 42-day-old *C. odorata* plants with different application rates of biochar. Legend: When the letters above the bar are different according to the multiple range Dunnet test, then it indicates that there are statistical differences at 95% between the application rates.

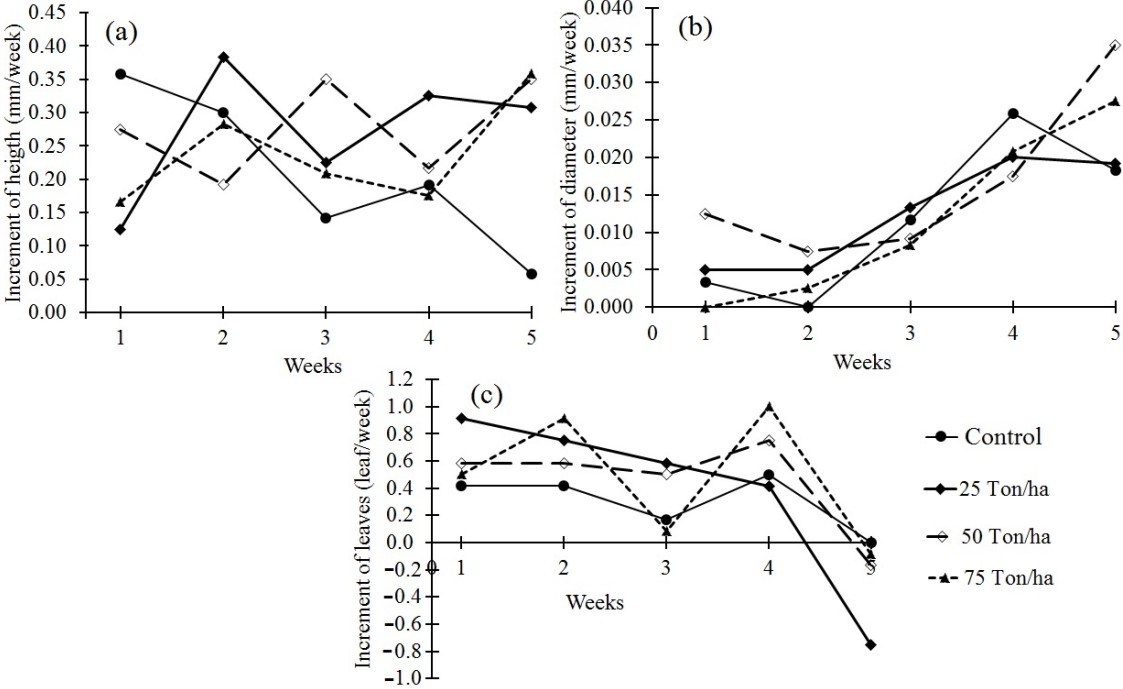

**Figure 3.** Variation of the increment of the height (**a**), diameter (**b**), and number of leaves (**c**) of plants of *C. odorata* with different application rates of biochar.

### 3.4. Biomass in Cedrela odorata Plant

To visualize the general behavior of the plants in the different biochar application quantities, Figure 4 shows the effect of the application rates on plant biomass. The application rate of 75 ton/ha biochar (Figure 4a) and the control (Figure 4d) showed poor root development and less foliage, which could hinder nutrient and water absorption in the plant. However, the plant from the soil with 50 and 75 ton/ha of biochar applied presented better development of the leaves and roots (Figure 4b,c) in these saplings.

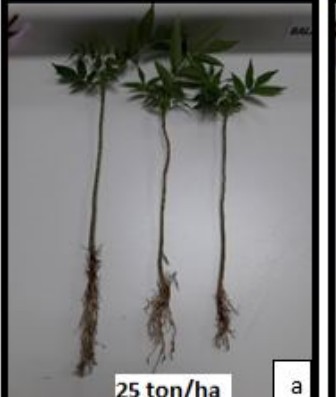 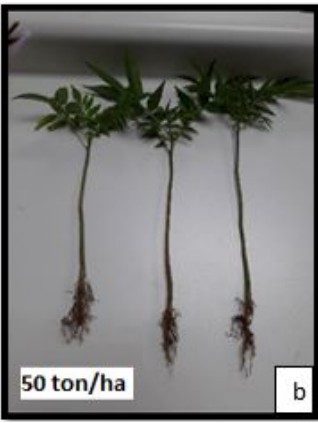 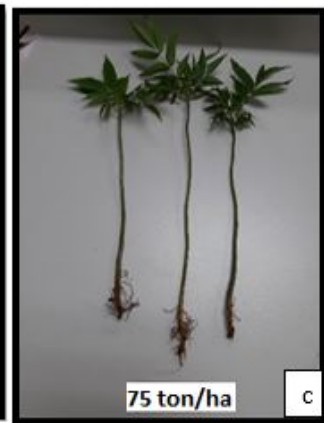 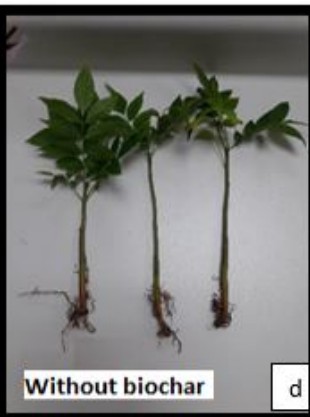

**Figure 4.** Plants of *C. odorata* with different application rates of biochar: (**a**) 25 ton/ha, (**b**) 50 ton/ha, (**c**) 75 ton/ha, (**d**) Control.

These observations are statistically supported by the results presented in Table 1 and Figure 1. However, for the application rates of 25 and 50 ton/ha of biochar, even though the plants showed better root and foliage development, no significant differences were observed (Table 2).

**Table 2.** Biomass production in dry conditions in *C. odorata* plants with different application rates of biochar.

| Application Rate (ton/ha) | Leaves (g) | Stem (g) | Root (g) | Total (g) |
|---|---|---|---|---|
| Control | 0.3607 | 0.6246 | 0.1208 | 1.1060 |
| 25 | 0.3179 | 0.5951 | 0.2341 | 1.1471 |
| 50 | 0.4353 | 0.5955 | 0.1622 | 1.1929 |
| 75 | 0.2215 *** | 0.4504 | 0.1219 | 0.7939 |

Legend: $p < 0.05$, there are significant differences ***.

The values of biomass for the different parts of 42-day-old plants are shown in Table 2. The biomass under dry conditions showed significant differences (Table 2); Dunnett's test showed that the application rate of 75 ton/ha as compared to the control had the lowest values among the three application rates evaluated. This performance can be supported with the help of Figure 4, where the thinnest plants and lowest leaf areas were observed in saplings supplied with the 75 ton/ha application rate. On the contrary, the application rate of 50 ton/ha presented a higher biomass of leaves, and total biomass.

### 3.5. Carbon, Hydrogen, Nitrogen, and Sulfur in the Plants

The percentage of C, H, N, and S and the C/N ratio and C/H ratios are shown in Table 3. As expected, the concentrations of N, C, and H were higher in the aerial part of the plants, because it is the photosynthetically active area. N presented significant differences in the leaves and stem for the application rate of 50 ton/ha of biochar with the highest values (approximately 54% in relation and 131% into the roots and stem, respectively). Additionally, for the application rate of 75 ton/ha, noteworthy differences were also observed for the values of leaves and root in relation to stem values: approximately 177% higher in leaves and roots. For the C content, the supply of 50 and 75 ton biochar/ha in the leaves presented higher percentages than in the roots and stems (which is approximately 5% higher). The H content showed considerable variations in the roots where the biochar was applied as 75 ton/ha with the highest value among the treatments at 3% higher. The S content was significantly higher (approximately 360% higher) in the stem for where the biochar was applied at rates of 25, 50, and 75 ton/ha.

**Table 3.** Carbon, hydrogen, nitrogen, and sulfur content in different parts of *C. odorata* with different application rates of biochar.

| Application Rate (ton/ha) | Part of Plant | N (%) | C (%) | H (%) | S (%) | C/N Ratio | C/H Ratio |
|---|---|---|---|---|---|---|---|
| Control | Leaves | 2.99 | 43.88 | 6.74 | 0.22 | 146.71 | 45.66 |
| | Root | 2.53 | 39.81 | 5.88 | 0.19 | 157.41 | 67.73 |
| | Stem | 1.21 | 40.36 | 6.57 | 0.06 | 333.38 *** | 61.44 |
| 25 | Leaves | 3.07 | 45.11 | 6.64 | 0.23 | 102.22 | 67.96 |
| | Root | 2.81 | 39.05 | 5.61 | 0.23 | 139.07 | 69.61 |
| | Stem | 1.25 | 42.06 | 6.57 | 0.06 *** | 337.34 *** | 64.05 *** |
| 50 | Leaves | 3.43 *** | 46.32 *** | 6.69 | 0.19 | 134.97 | 69.23 |
| | Root | 2.23 | 42.11 *** | 6.07 | 0.22 | 188.90 | 69.39 |
| | Stem | 1.48 *** | 42.33 *** | 6.47 | 0.06 *** | 287.03 *** | 65.47 *** |
| 75 | Leaves | 3.03 | 44.47 *** | 6.59 | 0.23 | 146.65 | 67.52 |
| | Root | 3.03 *** | 42.59 *** | 6.39 *** | 0.30 | 52.93 *** | 66.64 |
| | Stem | 1.09 *** | 42.69 *** | 6.68 | 0.05 *** | 393.27 *** | 63.88 *** |

Legend: $p < 0.05$, there are significant differences ***.

For the C/N ratio, significant differences were recorded in the stems where the biochar was supplied at the rate of 25, 75, and 50 ton/ha (over 150%). However, the application of biochar, i.e., 75 ton/ha, presented the lowest C/N ratio (which is approximately 130% lower). For the C/H ratio, only the application of biochar for any rate showed significant differences in the stems (which is approximately 5% lower).

## 4. Discussion

Biochar obtained from the *T. grandis* wood improved certain functions in the soil and its effects were evidenced in young saplings growing in a nursery. Its application showed a positive reaction in the growth of the height, diameter, and number of leaves (Figure 1) in nursery-raised saplings. On the contrary, the effect of biochar on the growth in the height and number of leaves at the rate of 25 ton/ha as compared to the control is clearly evident in Figure 2b,c. This result agrees with a study conducted on *Avena sativa* L, whcih showed that biochar application increases the plant height [28]. This effect was also found in *Pinus* sp. in Brazil, where a greater plant height was attained with a substrate prepared with the biochar of *Pinus* sp. produced at 400 °C [29].

The increase in the height, diameter, and number of leaves of the saplings of *C. odorata* can be attributed to the fact that biochar improves the physical environment and increases the water and nutrient retention in the soil [30]. These characteristics of the soil probably improve the root growth at the 25 and 50 ton/ha application rates (Figure 4).

Another possible benefit in plant growth is when the biochar is applied, the N content in the leaves was increased, which was shown by the 50 and 75 ton/ha application rates (Table 3). This increase was due to the improvement of plant nutrition by physical-chemical changes [31], and stimulated diametric growth (Figure 2b) of the individual saplings of *C. odorata*. In *Rosmarinus officinalis*, Liu Xu [32] showed the importance of biochar in the plant N content. This study also showed that when the substrate was prepared with chicken manure compost, a low N concentration was found, but when the compost of chicken manure was mixed with biochar, the N concentration increased significantly. Similarly, a high C/N ratio in the roots, as occurred in the saplings of *C. odorata* after the supply of 75 tons/ha in the present study (Table 3), suggests a slower activity in the roots [33]. However, this slower activity can be the attributed to the differences in the variables measured in relation to the control for the application rate of (75 ton/ha) of biochar.

Regarding the results presented in Table 3 and Figure 4, the availability of N in the soil and stems was closely related to the density of the root length; in addition, the N content allowed the amount and strength in the plant absorption rate to increase [34], which is

expressed in the saplings that were supplied with 25 and 50 tons/ha of biochar, with the highest growth in the diameter and height of the plant (Figure 2). This result agrees with the results presented by Wang et al. [35], where they mentioned that an application rate higher than 30% of biochar inhibited root development, compromising the physiological development of plants. Therefore, the present results show that the best performances were observed after the application rates of 25 and 50 tons/ha of *T. garndis* biochar to *C. odorata* saplings.

## 5. Conclusions

On the basis of the present results, it may be concluded that the application of *T. grandis* biochar shows positive effects on the growth (height, number of leaves, and diameter) of *C. odorata* saplings during nursery growth. The application of 25 and 50 ton/ha of biochar produced from *T. grandis* wood showed a significant increase in the diameter, height, and number of leaves of *C. odorata* saplings raised in a nursery. This is due to greater root development, which improves the nitrogen absorption in plants treated with *T. grandis* biochar. However, the concentration of biochar application gave variable results. Poor growth of *C. odorata* saplings was observed when the biochar was applied at the highest rate (i.e., 75 ton/ha). This suggests that the effects of biochar on plant growth and nutrition are highly complex and vary widely from species to species. Therefore, furthermore, further studies are warranted to confirm the results of the present investigation and to quantify the long-term benefits for improved and healthy development of saplings for plantation programs.

**Author Contributions:** Conceptualization, A.R.S., Y.B.V. and R.M.; methodology, A.R.S., Y.B.V. and R.M.; validation, A.R.S., Y.B.V. and R.M.; formal analysis, A.R.S., Y.B.V. and R.M.; investigation, A.R.S. and R.M.; resources, R.M.; writing—original draft preparation, A.R.S., Y.B.V. and R.M.; writing—review and editing, A.R.S. and R.M. SPS availability, Y.B.V. All authors have read and agreed to the published version of the manuscript.

**Funding:** This research received no external funding.

**Data Availability Statement:** No application.

**Acknowledgments:** The authors are grateful for the support of the Vicerrectoría de Investigación y Extensión of the Instituto Tecnológico de Costa Rica, who contributed the materials for this research.

**Conflicts of Interest:** The authors declare no conflict of interest.

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
