# Peer review of "Agronomic Effects of Tectona grandis Biochar from Wood Residues on the Growth of Young Cedrela odorata Plants in a Nursery"

_agronomy, doi:10.3390/agronomy11102079_

Round 1

Reviewer 1 Report

The presented work is interesting from the scientific and practical point of view regarding the use of biochar. However, before it is printed, it requires corrections as set out in the points below. 1. line 92 and 93 - Is the control understood as 0 tons / ha? If so, correct the sentence and use the word control instead of 0 throughout the text. 2. I also leave for consideration the inclusion of Figures 1b and 1c (it is described in the text and they do not contribute to the article). If you have urea you should write a chemical formula. 3. The work does not use subscripts in chemical formulas, eg line 137. 4. In table 4, please set the application rate starting with the control and ending with the highest applied dose. The current wording is confusing and different from the previous ones in the text. The same applies to Figure 5. Authors should opt for a uniform entry. 5. Conclusion needs to be redacted as well. It contains general information. The lack of a specific application

Author Response

Here word file

Reviewer 2 Report

The manuscript by Rodríguez-Solís et al. describes a simple study, that may have interest to the related community. The English must be revised and some aspects of the results must be changed. As it is now, this is a very descriptive study, so the authors must revise the writing to put a hypothesis-driven perspective. Please find my comments below:

L31 - delete 'of them'

L40 - Reference 6 is not in the right format

L43 - replace "of nursery plants"

L63 - replace 'to furniture' by 'in furniture'

L64 - presents a specific gravity lower than

L66 - and do not

L78 - of plant development

L79 - delete 'presented'

L80-82 - please change the aim and formulate a hypothesis

L91 - numbers below 10 must be in full: please revise the entire document

Figure 1 can be presented as supplementary data

L133 - delete 'the weight of'

L136 - chemical compounds must have the numbers in subscript

L156 - to obtain the dry weight.

L157 - revise and re-write

L177-180 - this sentence is too long and confusing, please re-write

Figure 2 and 4 - no statistical test was performed?

Figure 3 - please add statistical test performed in the legend of the figure and that different letters represent significant differences between treatments

L214 - The evaluation must be performed using the dry weight of the plants; please delete the text and results relative to "green" conditions

L234 - please be more quantitative in this section, what were the percentages of the significant differences registered between treatments?

L253 - please start with a context of the study

L256 - this result is in agreement with

L257 - increased plants height.

L258 - Pinus in italic

L260 - 'increase' instead of 'increasing'

L261 - replace the comma with 'and'

L262 - please re-write the sentence

L271 - Rosmarinus officinalis in italic

L275-278 - please re-write the sentence

L287 - these are not conclusions, just a summary of the results; please reflect on the importance of this study to the industry, environmental impact and possible economic outcomes of using biochar.

Author Response

here word file

Round 2

Reviewer 2 Report

The authors addressed all comments and I find the manuscript suitable for publication.